**Data Availability Statement:** All relevant data are within the manuscript and its Supporting Information files as well as submitted in Dryad data

# Post-COVID-19 syndrome among symptomatic COVID-19 patients: A prospective cohort study in a tertiary care center of Bangladesh

Reaz Mahmud[1]☯*, Md. Mujibur Rahman[2]☯¤a, Mohammad Aftab Rassel[1]‡, Farhana Binte Monayem[3]‡, S. K. Jakaria Been Sayeed[2]¤b‡, Md. Shahidul Islam[3]‡, Mohammed Monirul Islam[4]‡

1 Department of Neurology, Dhaka Medical College, Dhaka, Bangladesh, 2 Department of Medicine, Dhaka Medical College, Dhaka, Bangladesh, 3 Sarkari karmachari Hospital, Dhaka, Bangladesh, 4 Ministry of Health and Family Planning Welfare, Dhaka, Bangladesh

☯ These authors contributed equally to this work.
¤a Current address: Stroke Unit, National Institute of Neurosciences and Hospital, Dhaka, Bangladesh
¤b Current address: Department of Medicine, Bangabandhu Sheikh Mujib Medical University, Dhaka, Bangladesh
‡ These authors also contributed equally to this work.
* reazdmc22@yahoo.com

## Abstract

### Background

Post-coronavirus disease (COVID-19) syndrome includes persistence of symptoms beyond viral clearance and fresh development of symptoms or exaggeration of chronic diseases within a month after initial clinical and virological cure of the disease with a viral etiology. We aimed to determine the incidence, association, and risk factors associated with development of the post-COVID-19 syndrome.

### Methods

We conducted a prospective cohort study at Dhaka Medical College Hospital between June 01, 2020 and August 10, 2020. All the enrolled patients were followed up for a month after clinical improvement, which was defined according the World Health Organization and Bangladesh guidelines as normal body temperature for successive 3 days, significant improvement in respiratory symptoms (respiratory rate <25/breath/minute with no dyspnea), and oxygen saturation >93% without assisted oxygen inhalation.

### Findings

Among the 400 recruited patients, 355 patients were analyzed. In total, 46% patients developed post-COVID-19 symptoms, with post-viral fatigue being the most prevalent symptom in 70% cases. The post-COVID-19 syndrome was associated with female gender (relative risk [RR]: 1.2, 95% confidence interval [CI]: 1.02–1.48, p = 0.03), those who required a prolonged time for clinical improvement (p<0.001), and those showing COVID-19 positivity

repository. https://doi.org/10.5061/dryad.
m0cfxpp3g.

**Funding:** The author(s) received no specific
funding for this work.

**Competing interests:** The authors have declared
that no competing interests exist.

after 14 days (RR: 1.09, 95% CI: 1.00–1.19, p<0.001) of initial positivity. Patients with
severe COVID-19 at presentation developed post-COVID-19 syndrome (p = 0.02). Patients
with fever (RR: 1.5, 95% CI: 1.05–2.27, p = 0.03), cough (RR: 1.36, 95% CI: 1.02–1.81, p =
0.04), respiratory distress (RR: 1.3, 95% CI: 1.4–1.56, p = 0.001), and lethargy (RR: 1.2,
95% CI: 1.06–1.35, p = 0.003) as the presenting features were associated with the develop-
ment of the more susceptible to develop post COVID-19 syndrome than the others. Logistic
regression analysis revealed female sex, respiratory distress, lethargy, and long duration of
the disease as risk factors.

## Conclusion

Female sex, respiratory distress, lethargy, and long disease duration are critical risk factors
for the development of post-COVID-19 syndrome.

## Introduction

Since the first report of severe acute respiratory syndrome coronavirus 2 (SARS- CoV-2),
which causes coronavirus disease (COVID-19) on December 31 [1], the virus has dominated
the life of every person worldwide. The clinical presentation of COVID-19 ranges from asymp-
tomatic, mild symptomatic to fulminant and fatal cases. Severe cases of infection can lead to
serious complications including, pneumonia, acute respiratory distress syndrome (ARDS),
sepsis, multiple organ failure, blood clotting, myocarditis, acute myocardial infarction, acute
kidney injury, and other viral and bacterial infections that are not unique to coronavirus [2, 3].
COVID-19-associated death is possibly a result of pneumonia and hyperinflammation associ-
ated with cytokine storm syndrome [4].

The COVID-19 symptoms last for an average of 11.5±5.7 days [5]. However, a significant
proportion of patients have been found to remain unwell at post-discharge follow-ups [6]. In
the United Kingdom, a smartphone application-based study revealed the persistence of
COVID-19 symptoms in approximately 10% patients after 3 weeks of disease onset; in some
patients, the symptoms persisted for months [7]. The mechanism of this post-disease syn-
drome is unclear.

To address this issue, we need to define this condition first. There is no clear consensus on
the definition of post-COVID-19 syndrome. In this article, we have defined post-COVID-19
syndrome as (1) persistence of illness signs and symptoms (except fever, respiratory distress,
and hypoxia) after viral clearance (negative real time-polymerase chain reaction [RT-PCR]
results for COVID-19 at day 14 after initial positivity) or meeting the World Health Organiza-
tion (WHO) clinical criteria of improvement [8], including no fever for >3 days, improved
respiratory symptoms, pulmonary imaging showing obvious absorption of inflammation, and
no hospital care needed for any pathology or clinician assessment; (2) fresh development of
symptoms within a month after initial clinical and virological cure, the etiology of which is
postulated to be a viral infection (occurring after recovery); (3) exaggeration of previously
experienced chronic disease, such as migraine, mental disorder, bronchial asthma, and rheu-
matologic disorders, within a month after initial recovery from COVID-19.

Past experience with another coronavirus, severe acute respiratory syndrome coronavirus
(SARS-CoV), revealed post-viral fatigue syndrome/myalgic encephalomyelitis as the most
common symptom of the disease. As previously reported, the virus reaches the hypothalamus

via the olfactory pathway and disturbs its lymphatic drainage. This leads to the formation of pro-inflammatory cytokines, interleukins, and interferon gamma [9, 10] within the hypothalamus, which leads to the development of post-viral fatigability. Similarly, in COVID-19, the most common symptoms after acute COVID-19 are fatigue and dyspnea [11]. Diagnosis of post-viral fatigue [12] requires certain specific symptoms. It is most commonly accompanied by a substantial reduction or impairment in the ability to engage in pre-illness levels of occupational, educational, social, or personal activities that persist for >6 months. In many cases, there is new or definite onset (not lifelong) of profound fatigue, which is not the result of ongoing excessive exertion and is not substantially alleviated by rest. Further, post-exertional malaise and unrefreshing sleep are some of the common features. For further exploration of post-COVID-19 syndrome, it is necessary to have knowledge regarding the incidence, types, and risk factors of this syndrome. Therefore, this study aimed to determine the incidence, types of association, and risk factors for the development of post-COVID-19 syndrome in a cohort of patients with COVID-19.

## Materials and methods

This single-center prospective cohort study was performed to determine the extent of post-COVID-19 symptoms along with its risk factors in patients with COVID-19. The study was conducted in the COVID-19 unit of Dhaka Medical College Hospital from June 01, 2020 to August 10, 2020. Ethical approval was obtained from the Institutional Ethical Committee (ERC-DMC/ECC/2020/559).

### Participants

Patients with COVID-19 presented to the triage and inpatient department of Dhaka Medical College were screened for the study. The recruitment was limited to patients aged >18 years with confirmed SARS-CoV-2 positivity on RT-PCR. Asymptomatic or critical COVID-19 cases and patients unwilling to participate were excluded from the study. Informed written consent was obtained from all patients. The recruited patients were followed up for at least a month after clinical recovery and/or viral clearance. Consecutive patients were enrolled in the study. The sample size for this study was determined using the formula

$$\frac{z^2 pq}{d^2}$$

where z = 1.96 (at 95% confidence level); p = 50%, as the prevalence of post-COVID-19 syndrome is not known in Bangladesh; and q = (100−p) = 50. Here, d represents absolute error and was set at 5%. Therefore, the sample size, calculated as n = $(1.96)^2 \times 50 \times 50 / 5^2$, was 384 patients. A total of 400 patients were enrolled in the study.

### Study design

A case record form was constructed to collect baseline information of patients, such as demographics, clinical signs and symptoms, comorbidities, and oxygen saturation. Routine tests, including those for complete blood count, C-reactive protein, creatinine, random blood sugar, alanine aminotransferase, and D-dimer and chest X-ray, were advised on enrollment. RT-PCR testing for COVID-19 was performed 14 days after the initial positive test result for all the patients. A telephonic interview guide for the follow-up of patients after discharge was also developed. Patients were followed up via telecon for at least a month after recovery or hospital discharge. Clinical improvement was defined according to the WHO and Bangladesh guidelines [8, 13] as follows: normal body temperature for at least 3 days, significant improvement

in respiratory symptoms (respiratory rate <25breath/ minute and no dyspnea), oxygen saturation (SpO$_2$) >93% with no assistance for oxygen inhalation, and no hospital care needed for any pathology or clinician assessment. Respiratory distress was defined as shortness of breath, respiratory rate >25breath /min, or SpO$_2$ <93%, and mild disease was defined as symptoms of an upper respiratory tract viral infection, including mild fever, cough (dry), sore throat, nasal congestion, malaise, headache, muscle pain, anosmia, or malaise. Moderate disease was defined as respiratory symptoms such as cough and shortness of breath without signs of severe pneumonia. Severe disease was defined as severe dyspnea, tachypnea (>30 breaths/min), and hypoxia (SpO2 <90% in room air). Critical cases involved patients who developed ARDS or sepsis. These classifications were made according to the WHO and national guidelines of Bangladesh [8, 13]. The WHO has defined viral clearance as laboratory evidence of SARS-CoV-2 clearance in respiratory samples, i.e., two negative RT-PCR results using respiratory tract samples (nasopharynx and throat swabs), with a sampling interval of ≥24 h, after 14 days of initial positivity. However, due to limited testing facilities, we could perform RT-PCR only on day 14 after initial positivity for each patient. The criteria for post-COVID-19 syndrome considered in this research are described in the introduction of this manuscript. In our study, we have considered post-viral fatigue as symptoms reported in the literature and listed in the previous section along with any of the following: cognitive impairment and orthostatic intolerance. However, unlike previous reports, the criteria of its duration for 6 months was not considered in the present study.

## Procedure

Patients who met the inclusion criteria were enrolled in this study. Patients who required immediate hospital care were admitted. Routine and special investigations were performed according to the attending physician's advice. All patients received standard care of treatment as advised by the accompanying physicians. Patients were followed up every day and their conditions were recorded. RT-PCR for COVID-19 was performed on day 14 after initial positivity. After discharge, patients were followed up for at least a month via telecon using the telephone interview guide.

 Patients who did not require admission were sent home with appropriate treatment as recommended by the attending physician. They were advised to undergo routine investigations for their next visit. They were also followed up via telecon for at least a month after clinical recovery. Patients whose conditions deteriorated during the follow-up period were immediately advised for admission and were followed up similarly as those who received hospital care.

## Statistical analysis

A sample size of 400 patients would provide a power of at least 90% in the two-tailed test using a p-value of <0.05 to detect a 50% incidence of post-COVID-19 syndrome. Statistical Package for Social Sciences version 20 was used to analyze the data. Categorical variables are presented as n (%), normally distributed continuously are presented as mean (standard deviation [SD]), and skewed continuous variables are presented as median (interquartile range [IQR]). Statistical significance was set at p <0.05. For the comparison of variables, two groups were considered. Group 1 included patients who developed post-COVID-19 syndrome, and Group 2 included patients who did not develop post-COVID-19 syndrome. Categorical variables were compared using the chi-square test, and continuous variables were compared using an independent sample Student's t-test. Relative risk (RR) with a 95% confidence interval (CI) was calculated using crosstab analysis. The Mann–Whitney U test was used to compare skewed continuous variables. A binary logistic regression model was developed to assess the impact of

different variables on the likelihood of developing post-COVID-19 syndrome with the forward conditional method. Independent variables included in the model were age, sex, presenting features of COVID-19, duration of recovery, conversion to next level of severity, persistent positivity for the virus, comorbidities, and severity of illness.

## Results

Of the 486 patients who were screened and assessed for eligibility, 400 patients were enrolled in the study. In total, 42 patients were lost to follow-up and 3 patients died during follow-up. Hence, 355 patients completed the study (Fig 1).

The mean (SD) age of the study patients in was 39.8 (13.4) years. Most patients (60%) were younger than 40 years of age. The ratio of male and female patients was 1.4:1. Most patients presented fever (75%) and cough (62%), and a few (36%) patients showed signs of respiratory distress. Other important clinical features included anosmia (39%), hypoxia (30%), headache (20%), and lethargy (23%). Among the recruited patients, 62% patients had mild disease, 26% patients exhibited moderate disease, and 11% patients had severe disease. Some (27%) patients also had associated comorbidities (Table 1).

The incidence of post-COVID-19 syndrome was 46%. The median (IQR) interval between the recovery and development of post-COVID-19 symptoms was 7 (5–10.5) days. COVID-19 symptoms persisted beyond recovery in approximately 17% cases, whereas they developed after 7 days of recovery in 43% cases. In total, 105 (30%) patients shows at least one post-COVID-19 symptom, while 57 (16%) patients showed multiple symptoms. Post-viral fatigue was the most prevalent feature (117 [33%]). Other features included persistent cough (8.5%), post-exertional dyspnea (7%), headache (3.4%), vertigo (2.3%), and sleep-related disorders (5.9%) (Table 2).

Post-COVID-19 features were significantly higher among women (RR: 1.2, 95% CI: 1.02–1.48, p = 0.03), those who required a long time to achieve clinical improvement (p<0.001), and those showing COVID-19 positivity on RT-PCR after day 14 of initial positivity (RR: 1.09, 95% CI: 1.00–1.19, p<0.001). Additionally, patients with severe forms of the disease at presentation had a higher tendency to develop post-COVID-19 symptoms (p = 0.02).

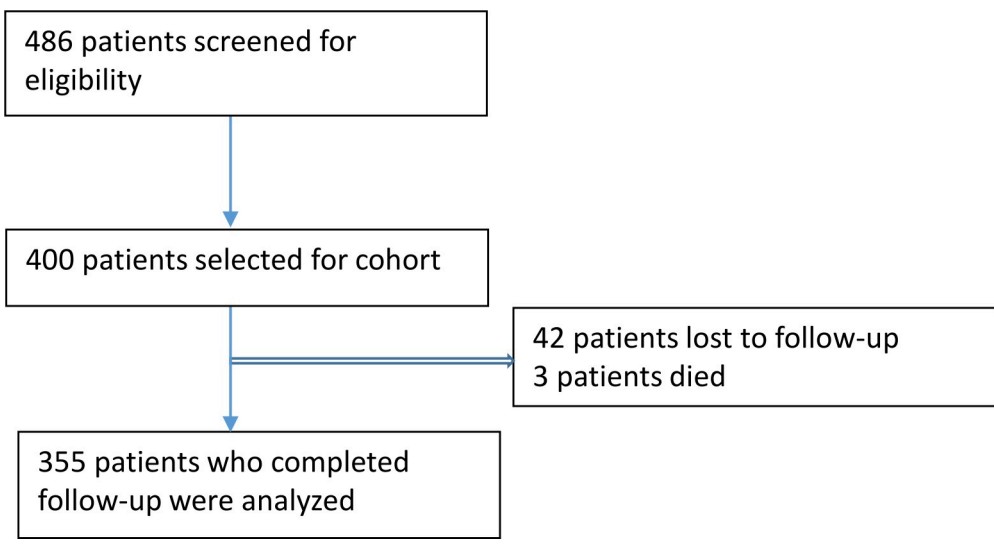

**Fig 1. Patient selection for this prospective cohort study.**

**Table 1. Baseline characteristics of COVID-19 patients with or without post-COVID-19 syndrome.**

| Variables | Total population n = 355 | Group 1[a] n = 162 | Group 2[b] n = 193 | p value | RR (95% CI) |
|---|---|---|---|---|---|
| Age (years), mean (SD) | 39.8 (13.4) | 40 (12.3) | 39.6 (14.3) | 0.81[c] | |
| Age <40 years, mean (SD) | 219 (61.7) | 100 (61.7) | 119 (61.7) | 0.65 | |
| Age = 40–60 years, mean (SD) | 107 (30.1) | 51 (31.5) | 56 (29) | | |
| Age >60 years, mean (SD) | 29 (8.2) | 11 (6.8) | 18 (9.3) | | |
| Sex (male), n (%) | 207 (58.3) | 84 (51.9) | 123 (63.7) | 0.03 | 1.2 (1.02–1.48) |
| Total duration of illness (days) median (IQR) | 12 (8–16) | 15 (10–20) | 10 (7–13) | <0.001[d] | |
| Fever, n (%) | 267 (75.2) | 131 (80.9) | 136 (70.5) | 0.03 | 1.5 (1.05–2.27) |
| Cough, n (%) | 224 (68.1) | 112 (69.1) | 112 (58) | 0.04 | 1.36 (1.02–1.81) |
| Respiratory distress[e], n (%) | 129(36.3) | 75(45.7) | 55(28.5) | 0.001 | 1.3 (1.14–1.56) |
| Running Nose, n (%) | 29 (8.2) | 11 (6.8) | 18 (9.3) | 0.44 | 0.97 (0.91–1.03) |
| Chest pain | 17(4.8) | 6(3.7) | 11(5.7) | 0.45 | 0.97 (0.94–1.03) |
| Sore throat, n (%) | 80 (22.5) | 28 (17.3) | 56 (26.9) | 0.03 | 0.88 (0.79–0.98) |
| Diarrhea, n (%) | 23 (6.5) | 11 (6.8) | 12 (6.2) | 0.83 | 1.01 (0.95–1.06) |
| Vomiting, n (%) | 17 (4.8) | 9 (5.6) | 8 (4.1) | 0.62 | 1.02 (0.97–1.06) |
| Anorexia, n (%) | 107 (30.1) | 56 (34.6) | 51 (26.4) | 0.12 | 1.12 (0.98–1.39) |
| Anosmia, n (%) | 138 (38.9) | 63 (38.9) | 75 (38.9) | 1 | 1 (0.85–1.18) |
| Headache, n (%) | 73 (20.6) | 37 (22.8) | 36 (11.7) | 0.36 | 1.05 (0.95–1.170 |
| Lethargy, n (%) | 81 (22.8) | 49 (30.2) | 32 (16.6) | 0.003 | 1.20 (1.06–1.35) |
| Body ache, n (%) | 66 (18.8) | 34 (21) | 32 (16.6) | 0.34 | 1.06 (0.95–1.17) |
| Persistent positivity[f] | 49 (13.8) | 29 (17.9) | 20 (10.4) | 0.04 | 1.09 (1.00–1.19) |
| Severity conversion[g] | 56 (15.8) | 28 (17.3) | 28 (14.5) | 0.56 | 1.03 (0.94–1.13) |
| Comorbidity | 97 (27.3) | 47 (29) | 50 (25.9) | 0.55 | 1.04 (0.96–1.19) |
| Hypertension, n (%) | 54 (15.2) | 21 (13) | 33 (17.1) | 0.30 | 0.95 (0.87–1.04) |
| Diabetes, n (%) | 49 (13.8) | 25 (15.4) | 24 (12.4) | 0.44 | 1.04 (0.95–1.13) |
| Severity grade[h] | | | | | |
| Mild, n (%) | 221 (62.3) | 90 (55.6) | 131 (67.3) | 0.02 | |
| Moderate, n (%) | 93 (26.2) | 54 (33.3) | 39 (20.2) | | |
| Severe n (%) | 41 (11.5) | 18 (11.1) | 23 (11.3) | | |

RR, relative risk; CI, confidence interval; SD, standard deviation; IQR, interquartile range.

[a] Group 1 patients who developed post-COVID-19 syndrome.

[b] Group 2 patients who did not develop post-COVID-19 syndrome.

[c] Independent sample t-test.

[d] Non-parametric test, Mann–Whitney U test.

[e] Shortness of breath, respiratory rate >25/min, or oxygen saturation <93%.

[f] Persistent positivity: patient who remained positive on the 14-day RT-PCR test after initial positivity.

[g] Patient experienced disease progression to the next level of severity during the follow-up from the initial presenting severity.

[h] Disease severity at presentation: mild symptoms of upper respiratory tract viral infection, including mild fever, cough (dry), sore throat, nasal congestion, malaise, headache, muscle pain, anosmia, or malaise; moderate respiratory symptoms such as cough and shortness of breath are present without signs of severe pneumonia (tachypnea >30 breaths/min and hypoxia: oxygen saturation <90% on room air).

**Table 2. Spectrum of post-COVID-19 symptoms.**

| Trait | Total patients n = 355, n (%) | Symptomatic patients n = 162, (%) |
|---|---|---|
| Post viral fatigue[a] | 117 (33) | 70.7 |
| Persistent cough[b] | 30 (8.5) | 18.3 |
| Insomnia[c] | 8 (2.3) | 4.9 |
| Circadian rhythm sleep disorders[d] | 14 (3.9) | 8.5 |
| Headache | 12 (3.4) | 7.3 |
| Vertigo | 8 (2.3) | 4.9 |
| Post-exertional dyspnea[e] | 25 (7) | 15.2 |
| Rash | 2 (0.6) | 1.2 |
| Pneumonia[f] | 2 (0.6) | 1.2 |
| Restless leg syndrome[g] | 2 (0.6) | 1.2 |
| Bradycardia | 2 (0.6) | 1.2 |
| Palpitation | 4 (1.4) | 2.4 |
| Anosmia | 7 (2) | 1.2 |
| Tinnitus | 1 (0.3) | 0.6 |
| Nasal blockade | 2 (0.3) | 1.2 |
| Chest pain | 3 (0.8) | 1.8 |
| Adjustment disorder[h] | 5 (1.4) | 3 |
| Arthralgia | 4 (1.4) | 4.8 |
| New-onset diabetes | 1 (0.3) | 0.6 |
| New-onset hypertension | 2 (0.6) | 1.2 |
| Non-ulcer Dyspepsia | 4 (1.4) | 4.8 |
| Excessive sweating | 4 (1.4) | 4.8 |
| Myalgia | 2 (0.6) | 1.2 |
| Burning feet | 1 (0.6) | 0.6 |
| Disturbance of memory[i] | 2 (0.6) | 1.2 |
| Precipitation of gout | 1 (0.3) | 0.6 |
| **Frequency of symptoms** | | |
| Single | 105 (29.6) | 65.2 |
| Multiple | 57 (16.1) | 34.8 |
| **Interval of symptom development from recovery** | | |
| From beginning | 27 (7.6) | 16.7 |
| <7 days | 64 (18.0) | 39.5 |
| >7 days | 71 (20) | 43.8 |

[a]A substantial reduction or impairment in the ability to engage in pre-illness levels of occupational, educational, social, or personal activities accompanied by profound fatigue.

[b]Coughing for >1 h or ≥3 coughing episodes in 24 h.

[c]Persistent difficulty with sleep initiation, duration, consolidation, or quality.

[d]Abnormalities in length, timing, and/or rigidity of the sleep–wake cycle relative to the day–night cycle.

[e]Perception of respiratory discomfort that occurs for an activity level that does not normally lead to breathing discomfort.

[f]New-onset bacterial pneumonia.

[g]Unpleasant sensation in the legs, causing overwhelming irresistible urge to move the legs, especially at bedtime.

[h]Emotional or behavioral symptoms occurring within 3 months of a stressor and lasting ≤6 months after the stressor or its consequences end.

[i]Pathological partial or complete loss of the ability to recall past experiences (retrograde amnesia) or to form new memories (anterograde amnesia).

Patients with fever (RR: 1.5, 95% CI: 1.05–2.27, p = 0.03), cough (RR: 1.36, 95% CI: 1.02–1.81, p = 0.04), respiratory distress (RR: 1.3, 95% CI: 1.4–1.56, p = 0.001), and lethargy (RR: 1.2, 95% CI: 1.06–1.35, p = 0.003) as the presenting features were more susceptible to develop post-COVID-19 syndrome compared to other presenting features. However, sore throat (RR: 10.88, 95% CI: 0.79–0.98, p = 0.03) was negatively associated with the development of post-COVID-19 syndrome (Table 1).

The logistic regression model explained 26% variation with 79% specificity. The variables made a unique statistically significant contribution to the model, as determined by forward conditional methods—female sex (odds ratio [OR]: 1.7, 95% CI: 1.1–2.8, p = 0.02), respiratory distress (OR: 0.21; 95% CI: 0.08–0.55, p = 0.001), lethargy (OR: 0.40, 95% CI: 0.22–0.70, p = 0.002), duration of illness (OR: 1.2, 95% CI: 1.1–1.2, p<0.001), and severity of illness (OR: 0.43, 95% CI: 0.20–0.93, p = 0.03) (Table 3). Some of the initial COVID-19 features overlap with the post-COVID-19 symptoms. However, the post-COVID symptoms have diverse presentations (Fig 2).

## Discussion

In this study involving 355 patients, the incidence of the post-COVID-19 syndrome was 46%, and most patients developed the symptoms after 7 days of initial recovery from the disease. The presentations varied widely; some patients had overlapping symptoms between COVID-19 and post-COVID-19 syndrome. Post-viral fatigue was the most common symptom, followed by persistent cough, exertional dyspnea, sleep disorders, adjustment disorders, and headache. Female sex, presenting features of respiratory distress, long recovery period, and disease severity were found to be risk factors for post-COVID-19 syndrome. Thus, this study revealed that the patients did not completely recover, even after apparent clinical recovery. The COVID-19 also caused long-term sequelae and distress in nearly half of the patients.

The demographics of patients with COVID-19 in this study varied from that of patients from Western countries. The most notable feature was patient age. In this study, 60% patients were aged <40 years and only 8% patients were aged >60 years. In a study from the USA, 31% patients were aged >65 years [14]. This is probably due to the sociocultural background of

**Table 3. Risk factors for post-COVID-19 syndrome (binary logistic regression analysis[a]).**

| Variables | Reference category | B[d] | SE[e] | Wald[f] | p value | Odd Ratio | 95% CI |
|---|---|---|---|---|---|---|---|
| Gender | Female | 0.56 | 0.24 | 5.0 | 0.02 | 1.7 | 1.1–2.8 |
| Respiratory distress[b] | Absence | −1.5 | 0.49 | 10.1 | 0.001 | 0.21 | 0.08–0.55 |
| Lethargy | Absence | −0.92 | 0.29 | 9.9 | 0.002 | 0.40 | 0.22–0.70 |
| Total duration of illness | | 0.15 | 0.03 | 33 | <0.001 | 1.2 | 1.1–1.2 |
| Severity[c] | Mild | −0.85 | 0.39 | 4.6 | 0.03 | 0.43 | 0.20–0.93 |
| Constant | | −0.005 | 1.1 | 0.00 | 0.99 | 0.99 | |

[a]Independent variables: sex, age, all COVID-19 symptoms, severity of COVID-19, severity conversion, persistent positivity, total duration of illness, patient suffering for <7 days and >14 days; Omnibus test of model coefficient, 0.00; Nagelkerke R square, 0.26; Hosmer–Lemeshow test, 0.83; Step 7, sensitivity 60%, specificity 79%.

[b]Shortness of breath, respiratory rate >25 breath/min, or oxygen saturation <93%.

[c]Disease severity at presentation: mild symptoms of upper respiratory tract viral infection, including mild fever, cough (dry), sore throat, nasal congestion, malaise, headache, muscle pain, anosmia, or malaise; moderate respiratory symptoms such as cough and shortness of breath without signs of severe pneumonia (tachypnea >30 breaths/min and hypoxia: SpO2 <90% on room air).

[d]This is the coefficient for the constant (also called the "intercept") in the null model.

[e]This is the standard error around the coefficient for the constant.

[f]The Wald chi-square value.

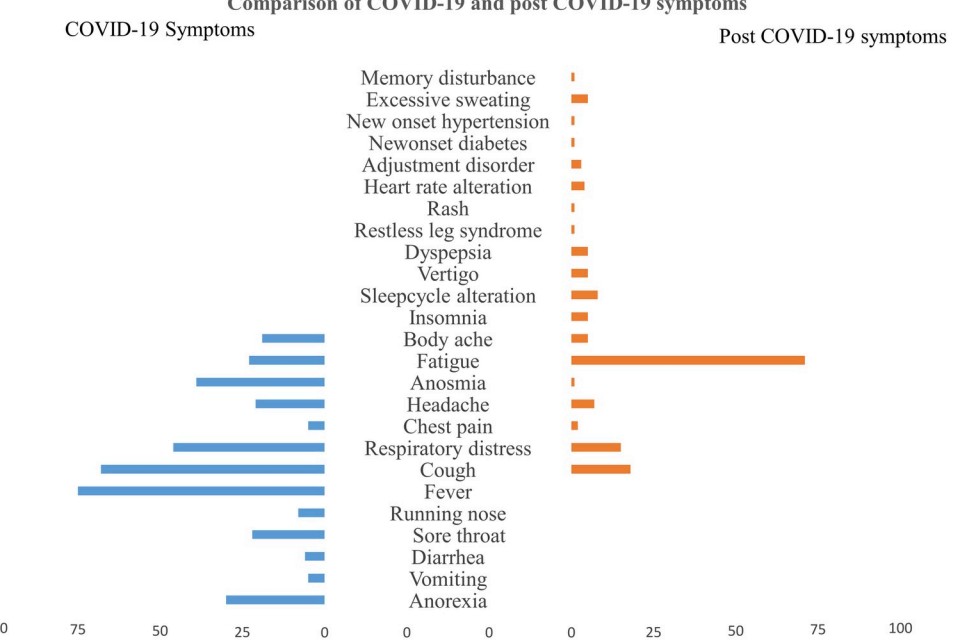

**Fig 2. Comparison of COVID-19 and post-COVID-19 symptoms.**

Bangladesh, where the proportion of the elderly population (5%) is lower than that in Western world (North America, 16%; Europe, 21%) [15]. In the present study, the proportion of men with COVID-19 was higher than that of women. Most patients presented fever, cough, anosmia, hypoxia, and lethargy at initial presentation of the disease. Previous studies have reported such a pattern [16, 17]. The median (IQR) for the duration of illness was 12 (8–16) days, which is consistent with that reported in a previous study [18].

There is no consensus regarding the persistence or fresh development of symptoms in the post-COVID-19 state, and the condition is not defined. Greenhalgh et al. [19] tried to define the conditions as "post-acute COVID-19" and "chronic COVID-19," which can extend beyond 3 weeks and 12 weeks, respectively, of the onset of first symptoms. This study was conducted among hospitalized patients, and approximately 10% patients showed post-COVID-19 symptoms. These patients required a long time to recover to meet the WHO criteria for recovery. Patients developing new symptoms or follow-up of mild COVID-19 symptoms were not considered in the study. Therefore, this study excluded a significant proportion of patients who had mild severity, did not need hospitalization, or developed new symptoms. Another study from Italy by Carfi et al. [6] followed-up the patients who met the WHO criteria for discontinuation of quarantine (no fever for consecutive 3 days, improvement in other symptoms, and two negative test results for SARS-CoV-2, 24 h apart. In approximately 87% cases, they found persistence of at least one symptom. However, the study sample was small, and a substantial number of patients received intensive care. Thus, the abovementioned studies are inadequate for explaining the post-COVID scenario as a whole. Moreover, a significant number of patients described in the studies were admitted to the intensive care unit. Patients admitted to the intensive care unit may develop symptoms such as executive dysfunction, anxiety, depression, and post-traumatic stress disorder due to post-intensive care syndrome [20]. If these symptoms are present in the post-COVID state among patients admitted to the intensive care

unit, it is very difficult to differentiate whether these symptoms are purely post-COVID-19 related or are outcomes of post-intensive care syndrome. To avoid such bias, we excluded critical patients requiring intensive care unit admission.

Thus, we attempted to define the condition by including patients with mild, moderate, and severe disease, and excluding critical patients admitted to the intensive care unit who have risk of developing post-intensive care syndrome. We found that about half of the patients developed new symptoms, had persistent mild COVID symptoms, or had exacerbated chronic diseases. Approximately 15% patients continued to have mild COVID-19 symptoms (excluding fever, cough, and respiratory distress) and 85% patients developed new symptoms. A study from the USA revealed that 35% patients did not return to their usual health status even after 3 weeks of COVID-19 positivity [21].

In this study, approximately 46% patients developed the post-COVID-19 syndrome. In another study, nearly 90% hospitalized patients who recovered from COVID-19 reported persistence of at least one symptom even after 2 months of discharge; 12.6% patients had no related symptoms, 32% patients had one or two symptoms, and 55% patients exhibited three or more symptoms [6]. In our study, 65% patients had at least one persistent symptom and 34% had multiple persistent symptoms.

Various post-COVID-19 symptoms have been reported in different studies. Post-COVID-19 symptoms can develop even in mild cases [6]. Most studies have reported fatigue, cough, respiratory distress, and headache as the dominant features [6, 7, 11, 19]. In our study, fatigue, persistent cough, exertional dyspnea, sleep disorders, and headache or vertigo were observed in 70%, 18%, 15%, 13%, and 12% cases, respectively. The reason for the dominance of fatigue was mostly unexplained. Viral infection-related immune system alterations may be the cause of fatigue [9, 10]. Cough and respiratory distress can be explained by persistent squeal lung damage. A recent study from China [22] reported decreased diffusion capacity for carbon monoxide in 25% patients 3 months after hospital discharge. In our study, a significant proportion of patients had sleep disturbances, including insomnia and circadian rhythm sleep disturbances. Previous experience with other SARS-CoV infections has revealed that involvement of the hypothalamus might be the reason for such symptoms [10]. We also found a large number of patients with adjustment disorders. Mental stress due to COVID-19 might have a role in developing adjustment disorders.

Risk factors related to post-COVID-19 syndrome were not identified in most previous studies [6, 7, 19, 21]. In this study, there was a significant association between post-COVID-19 syndrome and female sex, prolonged recovery, persistent positivity on RT-PCR after day 14 of the initial test, and moderate or severe illness at presentation. Fever, cough, respiratory distress, and lethargy were positively associated with the development of post-COVID-19 syndrome, but sore throat was negatively associated with the development of post-COVID-19 syndrome. The reason for this could not be explained. The following risk factors were identified in our study: female sex, respiratory distress, lethargy, long duration of illness, and moderate severity of the disease. All age groups had a similar susceptibility to develop post-COVID symptoms. In a study by Carfi A et al. [6], most COVID-19 symptoms persisted during the post-COVID-19 follow-up. However, in this study, we found an overlap of fatigue, cough, dyspnea, chest pain, headache, anosmia, and body ache (Fig 2). Diverse new symptoms including sleep disorder, adjustment disorder, memory disturbances, and restless leg syndrome that developed in the post-COVID-19 state also require attention.

This study included patients aged >18 years as it was difficult to comment about post-COVID-19 syndrome in younger patients. The demographics of the study patients were different from those of patients from Western countries. We did not find any association between age and the post-COVID-19 state; however, the findings might be different in other parts of the world.

This study has some limitations. It was a single-center study. Patients were followed up via a telephonic interview. Therefore, proper assessment of the patient's quality of life was not possible. Moreover, the follow-up period for patients was limited (1 month after disease onset), and asymptomatic or critical cases were excluded from the study. More representative findings could be obtained if all cases could be followed up for a longer period. Moreover, the effect size of most associated variables was small. A larger sample size is required to determine a strong association.

## Conclusions

Patients with COVID-19 require long-term follow-up even after recovery for observation and management of their post-COVID ailments. A comprehensive rehabilitation program is essential for such patients during hospitalization and discharge. During the ongoing COVID-19 pandemic, most health facilities are overloaded. Hence, arranging follow-up for patients can be a challenge. However, a significant population in the post-COVID state needs continuous monitoring. Female patients, patients presenting with respiratory distress, patients with lethargy, and patients with a disease for a prolonged duration require special attention in the post-COVID-19 state.

## Supporting information

**S1 Protocol. Post COVID syndrome.**
(DOCX)

**S1 File. Telephonic interview guide.**
(PDF)

**S1 Dataset. COVID 19 study post COVID.**
(XLS)

## Acknowledgments

We are grateful to every patient who gave their valuable consent for participation in this study; without their help, it would be impossible to conduct this study. We would like to thank Editage (www.editage.com) for English language editing.

## Author Contributions

**Conceptualization:** Reaz Mahmud, Md. Mujibur Rahman, Mohammad Aftab Rassel, Farhana Binte Monayem, S. K. Jakaria Been Sayeed, Md. Shahidul Islam, Mohammed Monirul Islam.

**Data curation:** Reaz Mahmud, Md. Mujibur Rahman, Mohammad Aftab Rassel, Farhana Binte Monayem, S. K. Jakaria Been Sayeed, Md. Shahidul Islam, Mohammed Monirul Islam.

**Formal analysis:** Reaz Mahmud, Mohammad Aftab Rassel, S. K. Jakaria Been Sayeed, Md. Shahidul Islam.

**Investigation:** Reaz Mahmud, Mohammad Aftab Rassel, Farhana Binte Monayem, S. K. Jakaria Been Sayeed, Md. Shahidul Islam, Mohammed Monirul Islam.

**Methodology:** Reaz Mahmud, Md. Mujibur Rahman, Mohammad Aftab Rassel, Farhana Binte Monayem, S. K. Jakaria Been Sayeed, Md. Shahidul Islam, Mohammed Monirul Islam.

**Project administration:** Reaz Mahmud, Md. Mujibur Rahman, S. K. Jakaria Been Sayeed.

**Resources:** Reaz Mahmud, Md. Mujibur Rahman, Mohammad Aftab Rassel.

**Software:** Reaz Mahmud.

**Supervision:** Reaz Mahmud, Md. Mujibur Rahman.

**Validation:** Reaz Mahmud, Md. Mujibur Rahman, Mohammad Aftab Rassel, Farhana Binte Monayem, S. K. Jakaria Been Sayeed, Md. Shahidul Islam, Mohammed Monirul Islam.

**Visualization:** Reaz Mahmud, Md. Mujibur Rahman, Mohammad Aftab Rassel, Farhana Binte Monayem, S. K. Jakaria Been Sayeed, Md. Shahidul Islam, Mohammed Monirul Islam.

**Writing – original draft:** Reaz Mahmud, Mohammad Aftab Rassel, Farhana Binte Monayem, S. K. Jakaria Been Sayeed, Mohammed Monirul Islam.

**Writing – review & editing:** Reaz Mahmud, Md. Mujibur Rahman, S. K. Jakaria Been Sayeed, Md. Shahidul Islam, Mohammed Monirul Islam.

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
