## [Decision Letter · Decision Letter 0]

2 Mar 2021

PONE-D-21-02914

Post COVID Syndrome among Symptomatic COVID-19 Patients: A Prospective Study in a Tertiary Care Center in Bangladesh

PLOS ONE

Dear Dr. Mahmud,

Thank you for submitting your manuscript to PLOS ONE. After careful consideration, we feel that it has merit but does not fully meet PLOS ONE’s publication criteria as it currently stands. Therefore, we invite you to submit a revised version of the manuscript that addresses the points raised during the review process.

We look forward to receiving your revised manuscript.

Kind regards,

Aleksandar R. Zivkovic

Academic Editor

PLOS ONE

Journal Requirements:

2) Please include captions for your Supporting Information files at the end of your manuscript, and update any in-text citations to match accordingly. Please see our Supporting Information guidelines for more information: http://journals.plos.org/plosone/s/supporting-information.

3) Please amend the manuscript submission data (via Edit Submission) to include author S.K.

Jakaria Been Sayeed.

4) Thank you for stating in the text of your manuscript "Informed consent was taken from every patients." Please state what type of consent you obtained (for instance, written or verbal, and if verbal, how it was documented and witnessed). Please also add all of this information to your ethics statement in the online submission form.

5) Please include a copy of the telephonic interview guide used in the study, in both the original language and English, as Supporting Information, or include a citation if it has been published previously.

6) We suggest you thoroughly copyedit your manuscript for language usage, spelling, and grammar. If you do not know anyone who can help you do this, you may wish to consider employing a professional scientific editing service.  

Reviewers' comments:

Reviewer's Responses to Questions

**Comments to the Author**

1. Is the manuscript technically sound, and do the data support the conclusions?

Reviewer #1: Partly

Reviewer #2: Yes

2. Has the statistical analysis been performed appropriately and rigorously? 

Reviewer #1: Yes

Reviewer #2: Yes

3. Have the authors made all data underlying the findings in their manuscript fully available?

Reviewer #1: Yes

Reviewer #2: Yes

4. Is the manuscript presented in an intelligible fashion and written in standard English?

Reviewer #1: No

Reviewer #2: No

5. Review Comments to the Author

Reviewer #1: Post COVID Syndrome among Symptomatic COVID-19 Patients: A Prospective Study

in a Tertiary Care Center in Bangladesh. This study presents a detailed description of incidence and risk factors of Post Covid sequelae. This is an interesting study, providing useful information for clinical practice.

Some points need to be clarified:

General:

The manuscript should go through a linguistic editing, as well as revision in accordance with academic writing guidelines

Abstract:

- Post COVID features are significantly higher among the female [RR, 95% CI, p 1.2, (1.02-1.48), 0.03], those who suffered for longer period (p= 0.00) and those who had prolonged positivity [RR, 95% CI, p; 1.09, (1.00-1.19), 0.00] for Covid 19. Please explain, it is not clear. The definitions seem to overlap, which is why p = 0.00

- Severity of the COVID had also positive association (p value=0.02). Please explain, it is not clear.

Main text:

- “the virus is dominating the life of every people of this universe.” please refer to our World only or add a reference about Covid people infection in other worlds.

- “To address this health problem and plan for future action.” This is very interesting but, this health problem has been already considered: in literature there are reviews about the Covid sequelae and rehabilitation of patients post-covid-19 infection. Please address this aspect in your article describing the status of art in this field. What do you mean for future action?

- Post COVID features are significantly higher among the female [RR, 95% CI, phi, p 1.2, (1.02-1.48), 0.12, 0.03], those who suffered for longer period (partial eta squared p value 0.18, 0.00), and those who had prolonged positivity [RR, 95% CI, phi, p; 1.09, (1.00-1.19), 0.11, 0.0] for COVID 19. The same as above. The definitions seem to overlap, which is why p = 0.00

- “It seems that they were the patient who required log time to recover to meet WHO criteria for recovery.” Not clear.

- “a new symptoms”. Correct this.

- “So the suffering of the COVID patients does not end with apparent clinical recovery. It also leaves long term sequels and sufferings for nearly half of the patients which also need to be addressed with proper attention.” Add a reference.

- “Mental agony related to having COVID infection might have a role.” What do you mean with mental agony?

- “More over intensive care unit patient may develop symptoms unrelated to COVID, due to post intensive care syndrome.” Which are these symptoms and how can you differentiate them from COVID symptoms, considering that covid is the cause of intensive care admission?

- “Those two study failed to identify the scenario as a whole.” Please don’t use the verb fail to.

- “COVID- 19 affected patients require long-term follow up even after recovery for observation and managing their ailments.” COVID-19 affected patients need a rehabilitation program during hospitalization and most of all at discharge. Please consider this aspect.

Reviewer #2: PONE-D-21-02914

Mahmud et al. report the prevalence of post-COVID-19 syndrome among relatively young patients in Bangladesh. Although this study was limited by single-centered design, fairly early time for evaluation, and more than 10% of loss-to-follow, it is still surprising to see a high prevalence of post-COVID-19 syndrome among young patients and its potential social impact in post-COVID-19 era. I would like to point out the following concerns:

Major comments:

1. While this study appears to be sound, the manuscript requires extensive elaboration on language to achieve clarity. In addition to the grammatical editing, the use of subheadings in the Methods and Results section will help to organize and improve the flow and readability of the manuscript.

2. For new symptom onset after the COVID-19, the data were not stratified further to determine if the symptoms were persistent following initial COVID-19, worsened after COVID-19 recovery, or occurred post-recovery. It would be valuable if the authors could delineate the prevalence more concisely.

3. The overall values of coefficient are low; thus, their interpretation is not clear. In contrast, Table 3 provides clear data, and this alone may be sufficient to report.

Minor comments:

1. In Table 3, if the purpose of the authors were to emphasize the independent risk factors for post-COVID-19 syndrome, it is more intuitive to use the mild group as a reference group fir severity.

2. What is the X axis of Figure 2? Is it the percentage of each symptoms among patients who developed the post-COVID-19 syndrome all COVID-19 patients?

3. I have trouble finding figure legends. Please include them in the revision.

4. Please edit Figure 1 as appropriate because eligible patients seem to increase from 352 to 355.

5. Please make sure to mention all the statistical methods used in the Methods section. For example, Mann-Whitney test is used in the results but does not appear in the Methods.

6. Please spell out RBS, SGPT considering the broad spectrum of the readers of this journal.

7. Please define “respiratory distress” in the Methods.

6. PLOS authors have the option to publish the peer review history of their article (what does this mean?). If published, this will include your full peer review and any attached files.

Reviewer #1: No

Reviewer #2: No

---

## [Author Response · Author response to Decision Letter 0]

19 Mar 2021

Response to Academic editor

1. PLOS ONE style-I have revised the manuscript according to PLOS ONE style.

2. Captions for Supporting Information files-Added at the end of the manuscript

3. The manuscript was edited by Editage. A copy was uploaded as supporting information. A clean copy of edited manuscript was uploaded as manuscript file.

4. Inclusion of S.K. Jakaria Been Sayeed- Included in the author list

5. About informed consent: Informed written consent was obtained. Mentioned in the manuscript( page 5 line 97-98)

6. Telephonic interview guide-added as supporting information.

7. The manuscript was edited by Editage. The certificate is added as supporting file

Thanks for reviewing the manuscript and giving the important comments about the manuscript.

Reviewer's Responses to Questions

Comments to the Author

1. Is the manuscript technically sound, and do the data support the conclusions?

Reviewer #1: Partly

Response: I have tried to correct the issues you raised in the subsequent section.

Reviewer #2: Yes

2. Has the statistical analysis been performed appropriately and rigorously?

Reviewer #1: Yes

Reviewer #2: Yes

Response: Thanks for your positive response.

3. Have the authors made all data underlying the findings in their manuscript fully available?

Reviewer #1: Yes

Reviewer #2: Yes

Response: Thanks for your positive response. I have also submitted the Data in the Dryad data repository to be published after acceptance of the manuscript. https://doi.org/10.5061/dryad.m0cfxpp3g

4. Is the manuscript presented in an intelligible fashion and written in standard English?

Reviewer #1: No

Reviewer #2: No

Response: Thanks for your critique. The manuscript is now edited by editage for its language and grammar. I think now you will find it standard. Certificate attached as supporting information

5. Review Comments to the Author

 Reviewer #1: Post COVID Syndrome among Symptomatic COVID-19 Patients: A Prospective Study in a Tertiary Care Center in Bangladesh. This study presents a detailed description of incidence and risk factors of Post Covid sequelae. This is an interesting study, providing useful information for clinical practice.

Response: Thanks for your appraisal

Some points need to be clarified:

General:

The manuscript should go through a linguistic editing, as well as revision in accordance with academic writing guidelines

Response: Revised by Editage during resubmission.

Abstract:

- Post COVID features are significantly higher among the female [RR, 95% CI, p 1.2, (1.02-1.48), 0.03], those who suffered for longer period (p= 0.00) and those who had prolonged positivity [RR, 95% CI, p; 1.09, (1.00-1.19), 0.00] for Covid 19. Please explain, it is not clear. The definitions seem to overlap, which is why p = 0.00

- Severity of the COVID had also positive association (p value=0.02). Please explain, it is not clear.

Response: It is now written in the following way to clarify-

The post-COVID-19 syndrome was associated with female gender (relative risk [RR]: 1.2, 95% confidence interval [CI]: 1.02–1.48, p=0.03), those who required a prolonged time for clinical improvement (p<0.001), and those showing COVID-19 positivity after 14 days (RR: 1.09, 95% CI: 1.00–1.19, p<0.001) of initial positivity. Patients with severe COVID-19 at presentation developed post-COVID-19 syndrome (p=0.02). (Page 2, line 30-34)

Main text:

- “the virus is dominating the life of every people of this universe.” please refer to our World only or add a reference about Covid people infection in other worlds.

Response: sorry for the mistake. The word universe is now replaced with world.

Since the first report of severe acute respiratory syndrome coronavirus 2 (SARS- CoV-2), which causes coronavirus disease (COVID-19) on December 31 [1], the virus has dominated the life of every person worldwide. (Page 3, line 44-46)

- “To address this health problem and plan for future action.” This is very interesting but, this health problem has been already considered: in literature there are reviews about the Covid sequelae and rehabilitation of patients post-covid-19 infection. Please address this aspect in your article describing the status of art in this field. What do you mean for future action?

Response: I have replaced the line with “For further exploration of post-COVID-19 syndrome, it is necessary to have knowledge regarding the incidence, types, and risk factors of this syndrome”.( Page-4, line 81-82)

- Post COVID features are significantly higher among the female [RR, 95% CI, phi, p 1.2, (1.02-1.48), 0.12, 0.03], those who suffered for longer period (partial eta squared p value 0.18, 0.00), and those who had prolonged positivity [RR, 95% CI, phi, p; 1.09, (1.00-1.19), 0.11, 0.0] for COVID 19. The same as above. The definitions seem to overlap, which is why p = 0.00

- “It seems that they were the patient who required log time to recover to meet WHO criteria for recovery.” Not clear.

Response: thank you for pointing this. I have corrected this as following

Post-COVID-19 features were significantly higher among women (RR: 1.2, 95% CI: 1.02–1.48, p=0.03), those who required a long time to achieve clinical improvement (p<0.001), and those showing COVID-19 positivity on RT-PCR after day 14 of initial positivity (RR: 1.09, 95% CI: 1.00–1.19, p<0.001). Additionally, patients with severe forms of the disease at presentation had a higher tendency to develop post-COVID-19 symptoms (p=0.02). (Page 14 line 225-229)

- “a new symptoms”. Correct this.

Response: Corrected

- “So the suffering of the COVID patients does not end with apparent clinical recovery. It also leaves long term sequels and sufferings for nearly half of the patients which also need to be addressed with proper attention.” Add a reference.

Response: this statement I have made according to my study findings. As the statement is creating confusion I have replaced it with following –

Thus, this study revealed that the patients did not completely recover, even after apparent clinical recovery. The COVID-19 also caused long-term sequelae and distress in nearly half of the patients. (Page 16, line 268-270)

- “Mental agony related to having COVID infection might have a role.” What do you mean with mental agony?

Response: As the term agony is creating confusion it have been replaced with mental stress.

Mental stress due to COVID-19 might have a role in developing adjustment disorders.(page 18, line 328-329)

- “More over intensive care unit patient may develop symptoms unrelated to COVID, due to post intensive care syndrome.” Which are these symptoms and how can you differentiate them from COVID symptoms, considering that covid is the cause of intensive care admission?

Impression: we had excluded the ICU patient from our study. As this line is creating confusion we rewrite it as

Thus, the abovementioned studies are inadequate for explaining the post-COVID scenario as a whole. Moreover, a significant number of patients described in the studies were admitted to the intensive care unit. Patients admitted to the intensive care unit may develop symptoms such as executive dysfunction, anxiety, depression, and post-traumatic stress disorder due to post-intensive care syndrome [20]. If these symptoms are present in the post-COVID state among patients admitted to the intensive care unit, it is very difficult to differentiate whether these symptoms are purely post-COVID-19 related or are outcomes of post-intensive care syndrome. To avoid such bias, we excluded critical patients requiring intensive care unit admission. (Page 17, line 294-301)

- “Those two study failed to identify the scenario as a whole.” Please don’t use the verb fail to.

Response: thanks for addressing this. I have replace the word.

Thus, the abovementioned studies are inadequate for explaining the post-COVID scenario as a whole. (page17, line 294)

- “COVID- 19 affected patients require long-term follow up even after recovery for observation and managing their ailments.” COVID-19 affected patients need a rehabilitation program during hospitalization and most of all at discharge. Please consider this aspect.

Response: Thank you for addressing this aspect. I have added this aspect in the text.

A comprehensive rehabilitation program is essential for such patients during hospitalization and discharge (page 20, line 357-358)

Reviewer #2: PONE-D-21-02914

Mahmud et al. report the prevalence of post-COVID-19 syndrome among relatively young patients in Bangladesh. Although this study was limited by single-centered design, fairly early time for evaluation, and more than 10% of loss-to-follow, it is still surprising to see a high prevalence of post-COVID-19 syndrome among young patients and its potential social impact in post-COVID-19 era. I would like to point out the following concerns:

Major comments:

1. While this study appears to be sound, the manuscript requires extensive elaboration on language to achieve clarity. In addition to the grammatical editing, the use of subheadings in the Methods and Results section will help to organize and improve the flow and readability of the manuscript.

Response: thank you for your concern and advice. I am very grateful that you reviewed the article with utmost importance. 

The article was edited by Eitage during resubmission for linguistic and grammatical error. In some context I have also made elaboration.

Subheadings added in the Methods and result section as per APA guideline. Some section were rewritten.

2. For new symptom onset after the COVID-19, the data were not stratified further to determine if the symptoms were persistent following initial COVID-19, worsened after COVID-19 recovery, or occurred post-recovery. It would be valuable if the authors could delineate the prevalence more concisely.

Response: It occurred post-recovery, to avoid confusion I have rewritten the line as

fresh development of symptoms (Page-2 line 18)

3. The overall values of coefficient are low; thus, their interpretation is not clear. In contrast, Table 3 provides clear data, and this alone may be sufficient to report.

Response: In revised manuscript it was omitted

Minor comments:

1. In Table 3, if the purpose of the authors were to emphasize the independent risk factors for post-COVID-19 syndrome, it is more intuitive to use the mild group as a reference group fir severity.

Response: I have corrected the table 3 according to your instruction

2. What is the X axis of Figure 2? Is it the percentage of each symptoms among patients who developed the post-COVID-19 syndrome all COVID-19 patients?

Response: added. It is the percentage of each symptoms among all COVID -19 patients on right side and among the patients with post-COVID syndrome on left side.

3. I have trouble finding figure legends. Please include them in the revision.

Responses: added 

4. Please edit Figure 1 as appropriate because eligible patients seem to increase from 352 to 355.

Response: Edited, error was in calculation

5. Please make sure to mention all the statistical methods used in the Methods section. For example, Mann-Whitney test is used in the results but does not appear in the Methods.

Response: Added in the revised manuscript

The Mann–Whitney U test was used to compare skewed continuous variables. (Page 8, line 162)

6. Please spell out RBS, SGPT considering the broad spectrum of the readers of this journal.

Response: Done in the revised manuscript.

7. Please define “respiratory distress” in the Methods.

Response: Definition added to the methods section

Respiratory distress was defined as shortness of breath, respiratory rate >25breath /min, or SpO2 <93 %,( page 6, line 119-120)

Thanks

Dr. Reaz Mahmud

Principal investigator

---

## [Editor Report · Decision Letter 1]

23 Mar 2021

Post-COVID-19 syndrome among symptomatic COVID-19 patients: A prospective cohort study in a tertiary care center of Bangladesh

PONE-D-21-02914R1

Dear Dr. Mahmud,

We’re pleased to inform you that your manuscript has been judged scientifically suitable for publication and will be formally accepted for publication once it meets all outstanding technical requirements.

Kind regards,

Aleksandar R. Zivkovic

Academic Editor

PLOS ONE

---

## [Editor Report · Acceptance letter]

30 Mar 2021

PONE-D-21-02914R1 

Post-COVID-19 syndrome among symptomatic COVID-19 patients: A prospective cohort study in a tertiary care center of Bangladesh 

Dear Dr. Mahmud:

I'm pleased to inform you that your manuscript has been deemed suitable for publication in PLOS ONE. Congratulations! Your manuscript is now with our production department. 

Kind regards, 

on behalf of

Dr. Aleksandar R. Zivkovic 

Academic Editor

PLOS ONE